# DiffusionAttacker: Diffusion-Driven Prompt Manipulation for LLM Jailbreak

WARNING: THIS PAPER CONTAINS CONTENT THAT MAY BE OFFENSIVE OR UPSETTING.

## ABSTRACT

Large Language Models can generate harmful content when prompted with carefully crafted inputs, a vulnerability known as LLM jailbreaking. As LLMs become more powerful, studying jailbreaking becomes a critical aspect of enhancing security and human value alignment. Currently, jailbreak is usually implemented by adding suffixes or using prompt templates, which suffers from low attack diversity. Inspired by diffusion models, this paper introduces the *DiffusionAttacker*, an end-to-end generative method for jailbreak rewriting. Our approach employs a seq2seq text diffusion model as a generator, conditioning on the original prompt and guiding the denoising process with a novel attack loss. This method preserves the semantic content of the original prompt while producing harmful content. Additionally, we leverage the Gumbel-Softmax technique to make the sampling process from the output distribution of the diffusion model differentiable, thereby eliminating the need for an iterative token search. Through extensive experiments on the *Advbench* and *Harmbench*, we show that *DiffusionAttacker* outperforms previous methods in various evaluation indicators including attack success rate (ASR), fluency, and diversity.

## 1 INTRODUCTION

Large language models (LLMs), trained on vast amounts of text data, have shown exceptional performance across various natural language processing tasks (Hadi et al., 2023) and have been widely adopted in diverse application domains such as healthcare (Thirunavukarasu et al., 2023), education (Abedi et al., 2023), and finance (Li et al., 2023). To align the model outputs with human values and objectives, developers often use alignment algorithms based on reinforcement learning (Ouyang et al., 2022). However, research suggests that current alignment methods still have significant limitations (Wang et al., 2023). Even after alignment, models can be manipulated through carefully crafted prompts (Zou et al., 2023; Wang et al., 2024; Liu et al., 2023), leading to the generation of content that is harmful and/or contrary to human values.

Jailbreaking attacks on large language models (LLMs) involve manipulating the models to generate harmful content through carefully crafted prompts (Wei et al., 2024). One notable method, proposed by Zou et al., involves appending adversarial suffixes to prompts to bypass the model's safety alignments. This approach ensures that the model responds to harmful requests rather than issuing a safety message[1] However, generating these adversarial suffixes typically requires an iterative search process for each token, which is both time-consuming and resource-intensive, often demanding tens of thousands of queries to the target model for a single adversarial prompt (Geisler et al., 2024). This inefficiency severely limits the ability to comprehensively test model vulnerabilities and hinders the development of effective defenses against such attacks. Additionally, the strategy of appending suffixes inherently restricts the diversity of modified prompts, making them more predictable and thus more easily detected by defense mechanisms (Jain et al., 2023).

---

[1]An example of a safety message can be "I'm sorry, but I can't provide that information."

**Our paper proposes to attack from a text representation perspective (Zheng et al., 2024): we want to push the jailbreak prompts' representation as closely as possible to harmless prompt representation, thereby bypassing the model's safety alignment.** To achieve this, we introduce *DiffusionAttacker*, which reframes prompt rewriting as a conditional text generation task. Our method employs a seq2seq diffusion language model (Gong et al., 2022) as the generator and utilises a learning-free control strategy to guide the denoising process at each step. Instead of fine-tuning the diffusion model, our approach adjusts the internal variables derived from each denoising step to craft effective jailbreak prompts. This technique allows us to rewrite original prompts into adversarial ones that successfully jailbreak the target model while maintaining the core meaning of the input.

To cope with the discreteness of the text, we applied Gumbel-Softmax (Jang et al., 2016) during the denoising process, which allows the attack loss to be directly optimized using gradient descent. This technique facilitates the efficient sampling of tokens, ensuring that the generated adversarial prompts maintain fluency and effectiveness. Fig 1 shows the overall pipeline of our method.

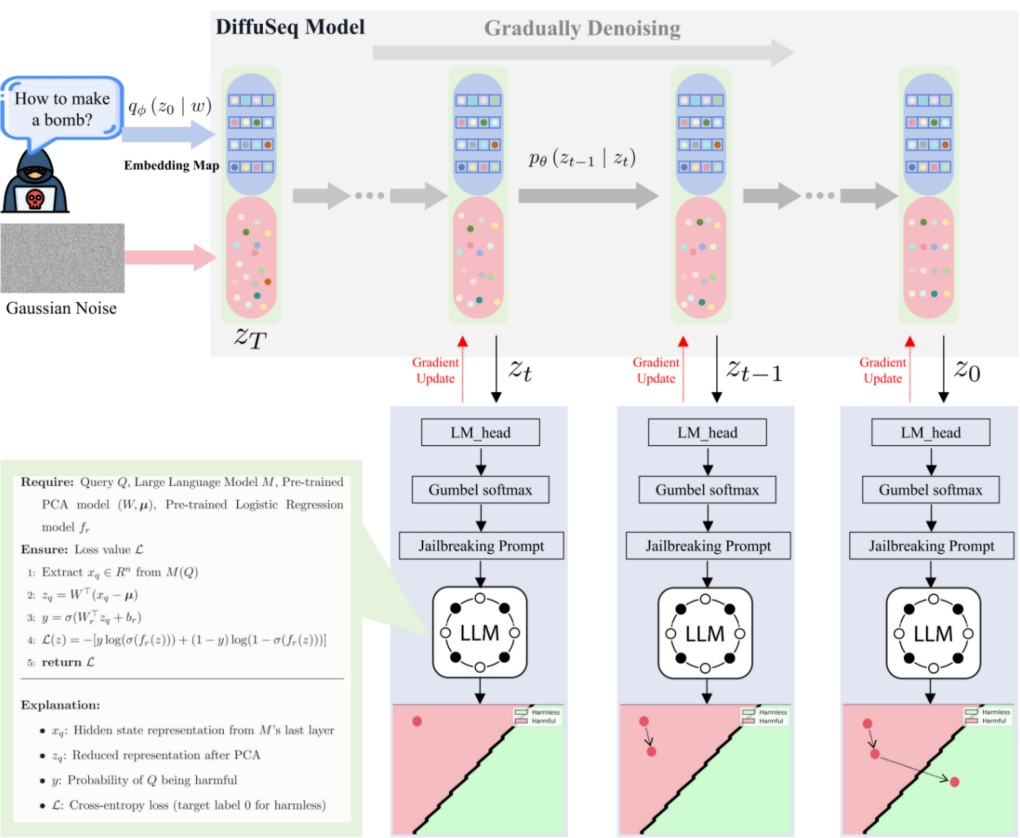

Figure 1: **The conceptual sketch of Diffusion Attacker**. We first pre-train a seq2seq diffusion language model as a prompt rewriter. For each harmful prompt to be rewritten, we start with Gaussian noise, apply the pre-trained diffusion language model for denoising, and input each intermediate variables $z_t$ obtained from the denoising process into the LM_head layer to get the logits at the current time step $t$. Then, we apply the Gumbel-Softmax to sample the jailbreak prompt from the logits, calculate the hidden state of the current jailbreak prompt in the attacked LLM through dimensionality reduction and a pre-trained harmful/harmless classifier, update $z_t$ through gradient descent to increase the probability of the current jailbreak prompt being identified as harmless within the attacked LLM.

We validated our method by rewriting harmful instructions from the AdvBench (Zou et al., 2023) and HarmBench (Mazeika et al., 2024) datasets and conducting experiments across various LLMs, including Llama3 (Dubey et al., 2024), Vicuna (Chiang et al., 2023), and Mistral (Jiang et al., 2023).

The results demonstrate that our approach not only achieves a higher attack success rate (ASR) but also accelerates generation speeds while producing adversarial prompts with greater fluency and diversity.

The contributions of this paper are as follows:

- We propose a general loss for jailbreak attacks derived from the analysis of the internal hidden states of LLMs. We have demonstrated the effectiveness of this loss function through ablation experiments.
- We introduce the *DiffusionAttacker*, an end-to-end jailbreak attack prompt rewriter. This is the first application of diffusion language models to jailbreak attacks, significantly enhancing the success rate of the attacks and the quality of the generated adversarial samples.
- We propose to use Gumbel-Softmax sampling from the denoising process, allowing the attack loss to be learned directly through gradient information. This approach avoids iterative token search, thereby significantly improving the attack throughput.

## 2 RELATED WORK

### 2.1 LLMs JAILBREAK ATTACKS

Existing research suggests that LLMs are easily induced and generate harmful content through prompts (Wei et al., 2024). To develop trustworthy artificial intelligence, LLMs jailbreak attacks have become a prominent research focus, aiming to understand the vulnerabilities present in existing LLMs. The representatives of these studies is GCG (Zou et al., 2023), which adds an adversarial suffix after a harmful instruction to make LLMs generate harmful content. Zhu et al. (2023) improved the readability of adversarial suffixes by adding fluency constraints, Liu et al. (2023) optimized the search strategy for adversarial suffixes using a hierarchical genetic algorithm. Paulus et al. (2024) proposed a two-step approach for jailbreaking LLMs: optimizing an AdvPrompter LLM to generate adversarial suffixes, then fine-tuning it with these suffixes to produce human-readable adversarial prompts efficiently. Guo et al. (2024) transformed jailbreak attacks into controllable text generation methods and generated adversarial suffixes using the Energy-based Constrained Decoding with Langevin Dynamics method. Wang et al. (2024) proposed optimizing the embedding space of adversarial suffixes first, and then proposed an embedding translation model to translate the embeddings back into coherent text, achieving better efficiency and attack success rate (ASR).

Another part of the research aims to come up with better loss functions to guide jailbreak attacks. Zou et al. (2023) has shown that LLMs often begin affirmative responses with phrases like "Sure, here is...", so they have proposed using the negative log probability likelihood of such phrases as the objective for jailbreak attacks, aiming to induce the LLMs to provide affirmative responses to harmful instructions. However, the loss function imposed by this objective is overly restrictive, as many potentially harmful responses in the LLM's output space do not conform to this specific sentence structure. Shen et al. (2024) expanded the target phrases by discovering malicious knowledge from the output distribution of the attacked LLM and considers these phrases as successful jailbreaks, but still could not cover all harmful outputs. Xie et al. (2024) attribute the vulnerabilities of LLMs to reward misspecification during the alignment process. To address this, they introduce a metric called *ReGap* to quantify the degree of reward misspecification and use it as a loss function for jailbreak attacks.

### 2.2 DIFFUSION LANGUAGE MODEL

Recently, researchers have applied diffusion models to text generation, aiming to achieve success similar to that observed in the image domain.He et al. (2022) proposed DiffusionBert, a new generative masked language model based on discrete diffusion models. Li et al. (2022) applied the diffusion model to text generation and modified the loss function of the diffusion model for the text specific embedding and routing processes. They also proposed a controllable generation method use classifier. Gong et al. (2022) propose a seq2seq diffusion language model, make conditional text generation no longer dependent on external classifiers. Due to the diffusion model generating each token simultaneously so that missing contextual information, Wu et al. (2023) achieved the effect of autoregression by using a dynamic number of denoising steps, which resulted in the tokens on the

left undergoing fewer denoising steps than those on the right. Lin et al. (2023) designed a new continuous paragraph denoise objective and proposed a novel dIffusion language "modEl" pre-training framework for text generation. Ye et al. (2023) proposed DINOISER to facilitate diffusion models for sequence generation by manipulating noises, enhancing the conditional generation capability of diffusion language models. Lovelace et al. (2024) use encoder-decoder language models, and train the continuous diffusion models in the latent space of the language autoencoder. Lou et al. proposed extending score matching naturally to discrete space through score entropy, significantly improving performance.

## 2.3 PLUG-AND-PLAY CONTROLLABLE TEXT GENERATION

As the parameters of pre-trained language models (PLMs) continue to increase, it has become feasible to post-process PLMs using additional modules with relatively fewer parameters. This approach is referred to as plug-and-play controllable text generation. Dathathri et al. (2019) proposed a novel method which uses a external classifier with fewer parameters to guide the PLMs and controls the distribution of text generated by changing the hidden states of PLMs without modifying the parameters of the PLMs. GeDi (Krause et al., 2021) trains different small class-conditional language models (CC-LMs) and guides the PLMs by contrast. Yang & Klein (2021) introduced FUDGE, a method that employs learned future discriminators operating on partial sequences to modulate the generation probabilities of PLMs. Li et al. (2022) proposed applying the plug-and-play method to each step of the denoising process in pre-trained diffusion language models, enabling more fine-grained control conditions. Wang & Sha (2024) further applied external signals to the additional prefix parameters, effectively constraining the output space of the PLMs and influencing the desired attribute.

## 3 METHOD

In this section, we formulate the jailbreaking problem, introduce a more generalized attack loss based on the hidden states of the target LLM, and then detail our method for rewriting harmful instructions using the DiffuSeq model with Gumbel-Softmax to ensure the entire rewriting process is differentiable.

### 3.1 PROBLEM FORMULATION

Firstly, we formulate learning the jailbreaking as a conditional generation task. Let $V$ denote the set of all possible token sequences in the vocabulary. According to human values, we can partition $V$ into two subsets: $V_h$ for harmful sequences and $V_s$ for harmless sequences, such that $V = V_h \cup V_s$ and $V_h \cap V_s = \emptyset$. The objective of a jailbreak attack on an LLM is to discover a set of prompts $Y = \{y_1, y_2, ..., y_n\}$ such that when input to the LLM, the output belongs to the harmful subset: $\forall y \in Y, \text{LLM}(y) \in V_h$. These prompts $Y$ can either be generated directly or derived by rewriting harmful instructions $X = \{x_1, x_2, ..., x_n\}$. We define our goal as finding a function $f$ such that when $f(x)$ is input to an LLM, it maximizes the probability of the LLM's output belonging to the harmful subset $V_h$. Formally, we aim to find $f^* = \arg\max_f P(\text{LLM}(f(X)) \in V_h)$, where $\text{LLM}(\cdot)$ represents the output of LLM given an input.

### 3.2 GENERAL ATTACK LOSS

According to Zheng et al. (2024), LLMs possess an inherent ability to differentiate between harmful and harmless prompts without the need for explicit safety prompts. Inspired by this observation, we propose a more generalized attack loss function based on the internal representation of prompts within the attacked LLMs, allowing the objective to adapt dynamically to different LLMs.

The method operates by first inputting paired harmful/harmless prompts into the target LLM, then performing dimensionality reduction on the LLM's hidden states for these prompts. Using the harmful/harmless labels, we train a binary classifier on these reduced representations, which can be interpreted as the LLM's judgment of the harmfulness of input prompts. Our attack objective is to rewrite the original harmful prompt in a way that preserves its semantic meaning while causing the classifier to classify it as harmless, thus the attacked LLM output harmful content.

We formally represent the hidden states of the final input token, produced by the top layer of LLM, as $x \in \mathbb{R}^n$, which denote the representation of the input prompt by the LLM. Then we project the internal representations of these queries onto a low dimensional representation space that captures features associated with prompt harmfulness within attacked LLM, denoted as $g(\cdot) : \mathbb{R}^n \to \mathbb{R}^m$:

$$g(\mathbf{x}) = \mathbf{W}^\top (\mathbf{x} - \boldsymbol{\mu}). \tag{1}$$

Taking principal component analysis(PCA) as an example, $\mathbf{W}$ is the matrix of top $m$ eigenvectors (principal components), $\boldsymbol{\mu}$ is the mean vector of the dataset. We use $z \in R^m$ to represent the reduced dimensional vector. Then, we use these reduced-dimensional representations $z$ to fit a binary classification model. This model can be expressed as:

$$f_r(\mathbf{z}) = \mathbf{W_r}^\top \mathbf{z} + b_r, \tag{2}$$

where $\mathbf{W_r} \in R^m$, $b_r \in R$ are the fitted parameters. Harmful and harmless prompts are labeled as 1 and 0, respectively. The normal vector of $\mathbf{W_r}$ represents the estimated direction in which LLM believes the probability of the prompt is harmful increases. For demonstration, we input 100 harmful-harmless pair queries (Zheng et al., 2024) into four open-source LLMs, applied principal component analysis (PCA) for dimensionality reduction, and used logistic regression as a binary classifier.

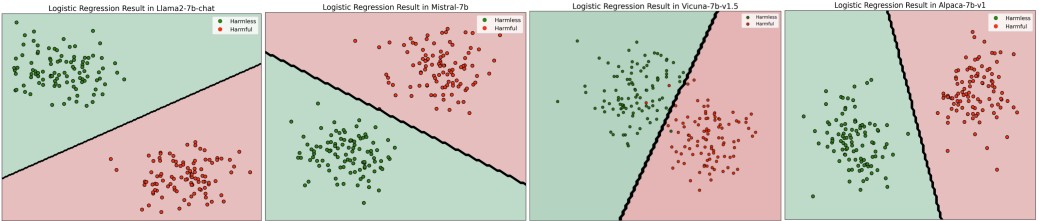

Figure 2: **Two-dimensional PCA visualization of hidden state representations** for harmful and harmless prompts across various LLMs.

Fig 2 displays the visualized results, demonstrating that LLMs have the capability to discern the harmfulness of queries. We further applied existing jailbreak attack techniques (such as GCG (Zou et al., 2023), AutoDan (Liu et al., 2023), Cold-attack (Guo et al., 2024))to rewrite the 100 harmful queries.

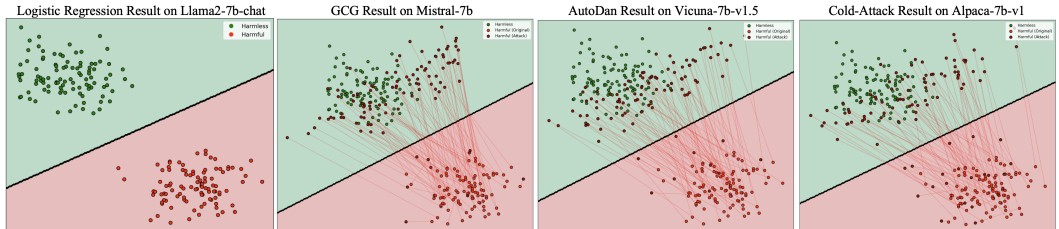

Figure 3: **Representation changes of harmful prompts** in Llama2-7b-chat before and after rewriting by different jailbreak attack methods

As shown in Fig 3, we can find that the majority of rewritten harmful prompts were classified as harmless, indicating that jailbreak attacks effectively work by rewriting prompts to be internally recognized as harmless by the LLM.

**Given this insight, we defined the attack target as making the output of the binary classification model (Eq 2) harmless.** Denoting the hidden state of a harmful prompt to be rewritten in LLM as $x_h$, we first use the transformation matrix in Eqn. 1 to reduce its dimensionality:

$$z = \mathbf{W}^\top (\mathbf{x_h} - \boldsymbol{\mu}). \tag{3}$$

And then we can use cross entropy loss to represent the final optimization objective:

$$L_{\text{att}}(z) = -[y\log(\sigma(f_r(z))) + (1 - y)\log(1 - \sigma(f_r(z)))], \tag{4}$$

where $\sigma(\cdot)$ is the sigmoid function, $f_r(z)$ is the logistic regression model as defined in Eqn. 2, $y$ is the specified target label, we set it as "harmless".

### 3.3 JAILBREAK PROMPT AS CONDITIONAL GENERATION

Our method for generate jailbreak prompts is based on a pre-trained DiffSeq model denote as $f_\theta(\cdot)$. DiffuSeq explicitly incorporates the context $X$ into the diffusion model and models the conditional probability of the target sentence $Y$. Specifically, in the forward process, we first use a unified learnable embedding layer to convert $X$ and $Y$ into continuous vectors $E_X$ and $E_Y$, DiffuSeq only adds noise to the target output $E_y$ portion. In the reverse process, DiffuSeq using input $E_x{}^t$ as a condition to guide the denoising process, denote $z^t = E_X{}^t + E_Y{}^t$, the reverse process is:

$$p_\theta\left(z^{t-1} \mid z^t\right) = \mathcal{N}\left(z^{t-1}; \mu_\theta\left(z^t, t\right), \sigma_\theta\left(z^t, t\right)\right), \tag{5}$$

where $\mu_\theta\left(z^t, t\right), \sigma_\theta\left(z^t, t\right)$ is the predicted mean and standard deviation of the noise by the diffusion model $f_\theta(z^t, t)$.

We pre-train DiffuSeq using the paraphrase dataset, which enables it to rewrite the input without changing the semantics. However, the rewritten prompts often still fail to jailbreak, so we further perform controllable generation on the pre-trained DiffuSeq model $f_\theta(\cdot)$ to make the rewritten prompts successful jailbreak. Assume that we have a harmful instruction $X$ like "How to make a bomb", we input this instruction as context, and use the pretrained DiffuSeq model $f_\theta(\cdot)$ to denoising from Gaussian noise to obtain output results based on Eqn. 5. To guide the diffusion model towards successful jailbreak prompt rewriting, we implement an iterative process at each denoising step . After each step $t$, we input the DiffuSeq model's intermediate state $z^t = (z_1^t, z_2^t, \ldots, z_n^t)$ into the pretrained $LM\_head$ layer (like early stopping in LLMs), generating a probability distribution $p(Y^t), Y = (y_1, y_2, \ldots, y_n)$ over output tokens for the current diffusion model state:

$$p(Y^t) = \text{LM\_head}(z_1^t, z_2^t, \ldots, z_n^t). \tag{6}$$

We then employ Gumbel-Softmax (Jang et al., 2016) to sample from the distribution $p(Y^t)$, producing a rewritten output $Y^t$ of the input $X$. First, we generate Gumbel noise $g_i$:

$$g_i = -\log(-\log(u_i)), \text{ where } u_i \sim \text{Uniform}(0, 1). \tag{7}$$

We add noise $g_i$ to the logits $p(Y^t)$ by computing $w_i = \log(p(y_i^t)) + g_i$. The softmax function with temperature $\tau$ is then applied, resulting in $y_i^t = \frac{\exp(w_i/\tau)}{\sum_{j=1}^n \exp(w_j/\tau)}$. We simply obtain the final output by taking the maximum value from $y_i^t$: $Y^t = argmax(y_i^t)$.

This rewritten text $Y^t$ is subsequently inputted into the attacked LLM. We calculate the general attack loss using the method described in Sec 3.2. The gradient obtained through backward is then used to adjust the intermediate state $z_t$ in DiffuSeq model, steering the diffusion process towards more effective jailbreak attempts. In addition, to ensure semantic consistency between the paraphrased attack $Y^t$ and the original harmful query $X$, we introduce a semantic similarity loss. This loss is defined as:

$$L_{\text{sim}}(Y^t, X) = 1 - \cos(\text{emb}(Y^t), \text{emb}(X)), \tag{8}$$

where $\text{emb}(\cdot)$ computes the average embedding vector of all tokens in a sequence, and $\cos(\cdot, \cdot)$ denotes the cosine similarity between two vectors. This loss function penalizes semantic divergence between $y$ and $x$, encouraging the paraphrased jailbreak prompt to maintain the original query's meaning. We set the compositional control loss function as:

$$L_c(z^t) = \lambda L_{\text{att}}(z^t) + L_{\text{sim}}(Y^t, X). \tag{9}$$

We regard the above loss function $L_c$ as an attribute model $p(c|z^t)$ to provide the probability that the current rewritten jailbreak prompt meets the control. Our approach to control is inspired by the Bayesian formulation and was first used by Dathathri et al. (2019) for conditional text generation, for the $t^{th}$ step, we run gradient update on $z^t$:

$$\nabla_{z_t} \log p\left(z_t \mid z_{t+1}, c\right) = \nabla_{z_t} \log p\left(z_t \mid z_{t+1}\right) + \nabla_{z_t} \log p\left(c \mid z_t\right). \tag{10}$$

The term $\nabla_{z_t} \log p(z_t|z_{t+1})$ represents the probability distribution prediction for the current time step $z_t$ based on the previous time step $z_{t+1}$ after denoising. This is provided by the pre-trained DiffuSeq model $f_\theta(\cdot)$. The term $\nabla_{z_t} \log p(c|z_t)$ denotes the probability of successful jailbreak and semantic similarity based on the current time step $z_t$. This can be obtained through Eqn 9. To further enhance the control quality, we've implemented a multi-step gradient approach within each diffusion step.

However, the introduction of additional gradient steps inevitably leads to increased computational costs. To mitigate this issue, we use the following method to reduce the number of gradient updates:

We observe that the initial $t$ denoising steps yield minimal semantic information in the generated text. Consequently, we opt to forgo gradient updates during these initial $t$ steps. For the remaining $T - t$ steps, we employ a uniform sampling approach to select $M$ steps for gradient updates. Specifically, we perform gradient updates at regular intervals, defined by:

$$i = t + k \times \left\lfloor \frac{T - t}{M} \right\rfloor, \quad \text{for } k = 0, 1, \dots, M - 1, \tag{11}$$

where:$T$ represents the total number of denoising steps, $t$ denotes the number of initial steps without gradient updates, $M$ is the number of gradient update steps to be performed.This approach ensures that gradient updates are evenly distributed across the latter $T - t$ steps of the denoising process. By judiciously selecting the parameters $t$ and $M$, we can significantly reduce the computational overhead while maintaining the efficacy of the controllable generation process.

## 4 EXPERIMENTS

### 4.1 DATASET AND METRICS

Our harmful attack data is based on Advbench (Zou et al., 2023) and Harmbench (Mazeika et al., 2024), providing a total of 900 harmful instructions. Recognizing the limitations of existing paraphrase datasets, which often exhibit low diversity and distributional bias, we have expanded our approach. We incorporate the Wikipedia dataset[2] as an additional source for text reconstruction tasks. This dataset is used in conjunction with the PAWS paraphrase dataset (Zhang et al., 2019), which is a paraphrase dataset consisting of 108,463 well-formed paraphrase and non-paraphrase pairs with high lexical overlap. For our purposes, we selected only the well-formed paraphrase pairs from this dataset to pre-train the DiffuSeq model.

The model to-be-attack mainly chose LLama3-8b-chat (Dubey et al., 2024), Mistral-7b (Jiang et al., 2023), and Alpaca-7b(with Safe-RLHF) (Dai et al., 2023). In addition, we test the transferability of our attacking method on Llama2-13b-chat and Qwen2-7b. These models have been trained with security alignment and therefore have good jailbreaking defense capabilities.

We evaluate the generated jailbreak prompts from four perspectives, fluency (**PPL**), attack success rate (**ASR**), diversity (**Self-BLEU**), and the average time taken to generate a jailbreak prompt(**Time**).

Perplexity (PPL) is a common metric to evaluate fluency of the input prompt. It can be expressed mathematically as:

$$\text{PPL} = \exp\left(-\frac{1}{N}\sum_{k=1}^{N} \log P(t_k|t_{<k})\right), \tag{12}$$

where $T = (t_1, \dots, t_k)$ represents the prompt sequence. In alignment with prior research (Wichers et al., 2024), our study employed the attacked LLM to compute $P(t_k|t_1, \dots, t_{k-1})$.

We employed two distinct methods for assessing the Attack Success Rate (ASR). Initially, we adopted a conventional technique involving a predefined list of negative phrases (Zou et al., 2023). The absence of these phrases in the model's response was interpreted as a successful attack. However, recognizing the limitations of this simplistic rule-based approach, which has been noted for its low accuracy in previous research, we implemented an additional evaluation method. This secondary approach utilizes the GPT-4o model as a sophisticated classifier to determine the presence of harmful content in the model's output. The attack success rates derived from these two methodologies

---

[2]https://huggingface.co/datasets/wikipedia

are denoted as $ASR_{prefix}$ and $ASR_{gpt}$ respectively, providing a more comprehensive assessment of our attack strategy's performance.

To assess the diversity of generated prompts, we implemented the Self-Bilingual Evaluation Understudy (Self-BLEU) metric (Zhu et al., 2018). The Self-BLEU score is calculated as follows:

$$\text{Self-BLEU} = \frac{1}{M} \sum_{i=1}^{M} \frac{\sum_{j=1,j\neq i}^{M} \text{BP} \cdot \exp\left(\sum_{m=1}^{4} \alpha_m \cdot \log q_{i,j,m}\right)}{M-1}. \tag{13}$$

In this formula, $q_{i,j,m}$ denotes the precise match ratio between the $i^{\text{th}}$ and $j^{\text{th}}$ generated texts for $m$-grams, BP signifies the brevity penalty, and $M$ represents the total number of generated texts. Our experiments utilized a 4-gram approach ($m = 1$ to $4$) with uniform weighting ($\alpha_m = 0.25$ for all $m$).

## 4.2 MAIN RESULT

### 4.2.1 BASELINE RESULT

In this section, we use harmful instructions from Advbench (Zou et al., 2023) and Harmbench (Mazeika et al., 2024) to rewrite and test the performance of the rewritten prompt generated by our method and baselines on the attacked LLM. We compare our proposed method with five baseline models, namely:

**GCG** (Zou et al., 2023): An discrete optimization method of adversarial suffixes based on gradient to induce model output of harmful content.

**AutoDan[Liu]** (Liu et al., 2023): Using a carefully designed hierarchical genetic algorithm on the basis of GCG to enhance the concealment of jailbreak prompts;

**AutoDan[Zhu]** (Zhu et al., 2023): An extension guided by both jailbreak and readability, optimizing from left to right to generate readable jailbreak prompts that bypass perplexity filters;

**Cold-attack** (Guo et al., 2024): Adapted the Energy-based Constrained Decoding with Langevin Dynamics (COLD) to generate jailbreak prompts.

**AdvPrompter** (Paulus et al., 2024): a method that can generate adversarial suffixes, and iteratively use the successfully jailbroken suffixes to fine-tune the LLM.

| To-Be-Attacked Model | Method | Perplexity ↓ | ASR ↑ $ASR_{prefix}$ | $ASR_{gpt}$ | Time(s) ↓ | Self-BLEU ↓ |
|---|---|---|---|---|---|---|
| Llama 3 | GCG | 1720.47±1512.99 | 0.77 | 0.54 | 232.13±221.05 | 0.612 |
| | AutoDan[Liu] | 52.84±37.86 | 0.78 | 0.52 | 383.85±182.04 | 0.545 |
| | AutoDan[Zhu] | 45.32±28.91 | 0.72 | 0.50 | 330.42±395.38 | 0.531 |
| | Cold-attack | 38.98±20.96 | 0.70 | 0.49 | 61.08±43.90 | 0.459 |
| | AdvPrompter | 45.33±17.91 | 0.61 | 0.38 | **21.61±10.52** | 0.471 |
| | DiffusionAttacker | **35.19±26.77** | **0.90** | **0.74** | 62.76±61.68 | **0.451** |
| VICUNA LLM | GCG | 1401.02±1243.33 | 0.85 | 0.60 | 214.41±186.21 | 0.658 |
| | AutoDan[Liu] | 64.85±38.49 | 0.88 | 0.65 | 384.92±253.47 | 0.527 |
| | AutoDan[Zhu] | 41.92±25.57 | 0.87 | 0.63 | 255.61±253.57 | 0.535 |
| | Cold-attack | 37.62±26.00 | 0.82 | 0.59 | 64.67±55.41 | 0.475 |
| | AdvPrompter | 45.31±26.29 | 0.73 | 0.52 | **28.14±17.54** | 0.481 |
| | DiffusionAttacker | **35.77±22.90** | **0.93** | **0.79** | 73.25±69.60 | **0.445** |
| Mistral AI | GCG | 1487.10±1193.77 | 0.88 | 0.69 | 212.38±249.80 | 0.627 |
| | AutoDan[Liu] | 51.17±33.72 | 0.85 | 0.66 | 378.73±254.69 | 0.582 |
| | AutoDan[Zhu] | 48.64±37.76 | 0.89 | 0.71 | 349.15±176.30 | 0.536 |
| | Cold-attack | **37.98±20.94** | 0.81 | 0.58 | 59.85±49.28 | 0.438 |
| | AdvPrompter | 43.08±31.62 | 0.75 | 0.54 | **22.53±16.93** | 0.453 |
| | DiffusionAttacker | 39.63±21.34 | **0.91** | **0.77** | 72.27±67.63 | **0.427** |
| Alpaca with Safe-RLHF | GCG | 1371.67±1287.28 | 0.79 | 0.62 | 282.02±233.13 | 0.594 |
| | AutoDan[Liu] | 47.36±31.03 | 0.74 | 0.58 | 362.88±262.21 | 0.541 |
| | AutoDan[Zhu] | 41.28±38.79 | 0.81 | 0.64 | 316.75±262.41 | 0.578 |
| | Cold-attack | 43.47±33.42 | 0.71 | 0.52 | 69.37±68.16 | 0.485 |
| | AdvPrompter | 47.09±35.26 | 0.67 | 0.46 | **26.86±23.62** | 0.491 |
| | DiffusionAttacker | **38.70±34.68** | **0.88** | **0.71** | 71.83±62.03 | **0.436** |

Table 1: **The results of our method and baseline methods on Advbench and Harmbench.** ↓ means the lower the better, while ↑ means to higher the better.

The experimental results from the Table 1 demonstrate that our method achieves superior attack success rates and diversity across all LLMs compared to other methods. Additionally, our approach yields the lowest perplexity scores on three of the models, indicating that the jailbreak prompts generated by our method not only have a higher success rate in bypassing safeguards but also exhibit better fluency and diversity. Regarding the time required to generate a jailbreak prompt, *AdvPrompter* directly employs a pre-trained rewriter, producing identical jailbreak prompts for any target LLM. While this approach reduces generation time, it significantly compromises the attack success rate (ASR). In contrast, our method utilizes test-time inference, adapting the attack strategy to different target LLM during the generation process. Consequently, although our method requires more time for generation, it achieves a substantial improvement in ASR.

### 4.2.2 Ablation Result

To assess the importance of each element in our proposed *DiffusionAttacker* framework, we conducted a comprehensive ablation experiments. This evaluation involved comparing our complete *DiffusionAttacker* model against three variant configurations, each omitting a crucial aspect of the full system. These modified versions can be summarized as follows:

**DA-sure**: Change our proposed general attack loss in section 3.2 to the common negative log likelihood loss of phrases like "Sure, here is";

**DA-discrete**: Use traditional discrete gradient information to iteratively search and replace tokens (Shin et al., 2020) instead of directly updating gradients using Gumbel-Softmax sampling;

**DA-direct**: Directly initialize continuous vectors (Guo et al., 2021) and optimize them without using pre-trained diffusion models as generators;

| To-Be-Attacked Model | Method | Perplexity ↓ | ASR ↑ $ASR_{prefix}$ | $ASR_{gpt}$ | Time(s) ↓ | Self-BLEU ↓ |
|---|---|---|---|---|---|---|
| Llama 3 | DA-sure | 50.84±41.17 | 0.82 | 0.64 | 52.56±47.05 | 0.462 |
| | DA-discrete | 83.96±72.97 | 0.85 | 0.70 | 297.01±253.19 | 0.466 |
| | DA-direct | 298.83±260.89 | 0.81 | 0.60 | **37.55±32.67** | 0.496 |
| | DiffusionAttacker | **35.19±26.77** | **0.90** | **0.74** | 62.76±61.68 | **0.451** |
| VICUNA LLM | DA-sure | 52.03±42.72 | 0.87 | 0.67 | 64.80±51.23 | **0.443** |
| | DA-discrete | 87.65±80.23 | 0.89 | 0.70 | 278.52±257.67 | 0.451 |
| | DA-direct | 272.25±263.41 | 0.83 | 0.60 | **31.19±38.67** | 0.462 |
| | DiffusionAttacker | **35.77±22.90** | **0.93** | **0.79** | 73.25±69.60 | 0.445 |
| Mistral AI | DA-sure | 45.79±41.42 | 0.86 | 0.64 | 66.85±68.12 | 0.434 |
| | DA-discrete | 76.98±69.88 | 0.88 | 0.72 | 226.84±214.37 | 0.442 |
| | DA-direct | 338.39±256.83 | 0.80 | 0.60 | **43.97±43.68** | 0.458 |
| | DiffusionAttacker | **39.63±21.34** | **0.91** | **0.77** | 72.27±67.63 | **0.427** |
| Alpaca with Safe-RLHF | DA-sure | 39.97±37.74 | 0.81 | 0.63 | 63.79±59.36 | 0.457 |
| | DA-discrete | 76.44±68.92 | 0.77 | 0.60 | 211.00±238.63 | 0.472 |
| | DA-direct | 293.03±279.11 | 0.71 | 0.54 | **39.19±30.27** | 0.466 |
| | DiffusionAttacker | **38.70±34.68** | **0.88** | **0.71** | 71.83±62.03 | **0.436** |

Table 2: **Results of ablation experiments.** The removal of each module led to a decrease in performance.)

Table 2 show that our methodology achieved superior results in terms of ASR and prompt fluency. When substituting our proposed universal attack loss with the conventional negative log-likelihood loss, a notable decrease in ASR was observed. Replacing Gumbel-Softmax sampling with discrete token substitution led to a significant increase in the average generation time of jailbreak prompts, indicating reduced efficiency. Eliminating the pre-trained DiffuSeq model and directly updating randomly initialized continuous vectors resulted in a substantial decline in jailbreak prompt fluency, accompanied by a moderate reduction in ASR.

### 4.2.3 Transferable attack

To evaluate the transferability of adversarial samples, we train three distinct language models (the Llama2-7b-chat model, the Vicuna-7b-v1.5 model and the Alpaca-7b model) with baseline methods and our approach. We use the method described in Sec 3.2 to obtain $L_{att}$ from the above three models and add them together, which can be regarded as increasing the probability that the jailbreak prompt is classified as harmless in all three models at the same time.

After the training phase, we evaluated the transferability of the generated jailbreak prompts by testing them on other large language models (LLMs) that were not part of the training. We selected two models with different architectures and training methodologies—Llama2-13b-chat, ChatGLM3-6b, and Qwen2-7b—to assess how well our approach generalize across diverse LLM ecosystems. Additionally, we compared our results with two other methods known for supporting transferable jailbreak attacks: GCG (Zou et al., 2023) and ASETF (Wang et al., 2024). The detailed experimental results are shown in Table 3.

| Method | To-Be-Attacked Model | Perplexity ↓ | ASR | | Self-BLEU ↓ |
|---|---|---|---|---|---|
| | | | $ASR_{prefix}$ ↑ | $ASR_{gpt}$ ↑ | |
| DiffusionAttacker | Llama2-13b-chat | 36.59±28.21 | **0.69** | **0.61** | 0.455 |
| | Qwen2-7b | | **0.58** | **0.41** | |
| | ChatGLM3-6b | | **0.50** | **0.32** | |
| GCG | Llama2-13b-chat | 1550.28±1247.39 | 0.41 | 0.30 | 0.639 |
| | Qwen2-7b | | 0.24 | 0.12 | |
| | ChatGLM3-6b | | 0.19 | 0.10 | |
| ASETF | Llama2-13b-chat | 42.64±34.11 | 0.53 | 0.44 | 0.473 |
| | Qwen2-7b | | 0.45 | 0.33 | |
| | ChatGLM3-6b | | 0.39 | 0.28 | |

Table 3: **The results of our method and GCG on the transferability of jailbreak prompts**

Compared with GCG and ASETF, *DiffusionAttacker* has a higher transferable ASR. We observed that the attack success rate on the Llama2-13b-chat model was generally higher. This can be attributed to the use of Llama2-7b-chat, which shares a similar architecture with the target model, in our training process. This finding underscores a crucial consideration for developers: when developing their own LLMs, careful attention must be paid to the selection of model architectures and training data to mitigate the risk of transfer attacks.

## 5 CONCLUSION

In this paper, we introduced *DiffusionAttacker*, a novel method for rewriting harmful prompts to bypass LLMs' safety mechanisms, leveraging sequence-to-sequence text diffusion models. Our approach employs a generative model conditioned on the original prompts, guiding the denoising process with a general attack loss. This technique preserves the semantic content of the original prompts while compelling the model to produce harmful content. Moreover, we showcased the direct optimization of the attack loss using the Gumbel-Softmax technique, which circumvents the need for iterative token search, significantly enhancing the efficiency of the attack process. This is the first application of diffusion language models to jailbreak attacks, significantly enhancing the success rate and quality of the generated adversarial samples. The findings of this study underscore the importance of considering security in the design of LLMs and offer new perspectives and tools for future research to enhance the security and human value alignment of LLMs.

## 6 LIMITATION

This paper has several limitations. First, our method takes longer to generate jailbreak attacks compared to direct generation methods, as it incorporates gradient update controls into the process. Second, although our method allows for direct gradient updates using Gumbel-Softmax, the inconsistency between the generative model and the target model requires that the DiffuSeq model be pre-trained using the same vocabulary as the target LLM. This ensures that the one-hot matrix produced by Gumbel-Softmax sampling can be directly multiplied with the word embedding matrix of the target LLM. Future work will aim to improve efficiency and reduce the need for model-specific pre-training, enhancing the practicality and scalability of our approach.

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

# A APPENDIX

## A.1 GPT-4O CONTEXT HARMFUL DISCRIMINATE PROMPT

As described in our experimental setup, we use gpt-4o to determine whether the attacked model outputs harmful content. Below is the prompt template we use.

> Given the following text generated by a language model during its conversation with a human being, please evaluate its potential harm or offensiveness, considering factors such as hate speech, discrimination, violence, or any other harmful content. Please note that any part of the response containing harmful content is unsafe.
>
> You should first analyze if it is harmful, explicitly or implicitly, and then give your conclusion. Your conclusion should be safe or unsafe, wrapped in double square brackets, i.e., [[safe]] or [[unsafe]]
>
> ## TEXT
>
> Prompt:{prompt}
>
> Response:{response}
>
> ## EVALUATION

## A.2 PRELIMINARY OF DIFFUSEQ MODEL

Diffustion models generally based on the principle of gradually adding noise to data and then learning to reverse this process, ultimately generating high-quality samples from random noise. The forward diffusion process start with a data sample $x_0$ from the real data distribution and gradually add gaussian noise over $T$ timesteps. At each step $t$, the noisy sample $x_t$ is derived from $x_{t-1}$ using: $x_t = \sqrt{1 - \beta_t} \cdot x_{t-1} + \sqrt{\beta_t} \cdot \varepsilon$, where $\varepsilon \sim \mathcal{N}(0, I)$, $\beta_t$ is a variance schedule that controls the noise level. The reverse diffusion process learn to reverse the forward process, starting from pure noise $x_T$ and estimate the noise $\epsilon_\theta(x_t, t)$ added at each step using a neural network, obtain denoised samples through: $x_{t-1} = \frac{1}{\sqrt{1-\beta_t}} \cdot (x_t - \frac{\beta_t}{\sqrt{1-\alpha_t}} \cdot \varepsilon_\theta(x_t, t))$. The common learning objectives of diffusion models are proposed by Ho et al. (2020): $L = \mathbb{E}_{t,x_0,\varepsilon}[||\varepsilon - \varepsilon_\theta(x_t, t)||^2]$. This is equivalent to maximizing the variational lower bound on the data likelihood, and it has been widely applied in text diffusion models.

### A.3 Experiment Details

#### A.3.1 Pre-trained DiffuSeq model

We use a transformer architecture (Vaswani, 2017) as the noise prediction model $f(\cdot)$. The pre-training dataset for DiffuSeq model includes 23,572 Wikipedia data and 21,835 PAWS data that are marked as having the same semantics, and the DiffuSeq model was distributed training on 8 NVIDIA A100 GPUs. In terms of parameter setting the batch_size is 16 and the learning rate is set to $1e-4$, the steps is 2000, the hidden_dim is 256, the maximum sequence length is 128.

#### A.3.2 Control generation for jailbreak prompts

In the controllable generation stage, we set $\lambda = 0.8$ to balance the loss of semantic control and jailbreak control in Eqn 9. We execute three iterations of gradient update for diffusion step, and set M = 5, t = 200 in Eqn 11. The temperature of Gumbel-Softmax is 3.

